# Potent Cytotoxicity of Novel L-Shaped Ortho-Quinone Analogs through Inducing Apoptosis

**DOI:** 10.3390/molecules24224138

**Published:** 2019-11-15

**Authors:** Sheng-You Li, Ze-Kun Sun, Xue-Yi Zeng, Yue Zhang, Meng-Ling Wang, Sheng-Cao Hu, Jun-Rong Song, Jun Luo, Chao Chen, Heng Luo, Wei-Dong Pan

**Affiliations:** 1College of Pharmacy, Guizhou University, Huaxi Avenue South, Guiyang 550025, China; 13118517665@163.com; 2State Key Laboratory of Functions and Applications of Medicinal Plants, Guizhou Medical University, 3491 Baijin Road, Guiyang 550014, China; xueyizeng@126.com (X.-Y.Z.); menglingwang@yahoo.com (M.-L.W.); 18275365116@163.com (J.-R.S.);; 3School of Medicine, Guizhou University, Huaxi Avenue South, Guiyang 550025, China; zekunsun@163.com; 4The Key Laboratory of Chemistry for Natural Products of Guizhou Province and Chinese Academy of Sciences, 3491 Baijin Road, Guiyang 550014, China; hushengcao0221@163.com; 5College of Agriculture, Guizhou University, Huaxi Avenue South, Guiyang 550025, China; zhangyueinsect@163.com

**Keywords:** ortho-quinones, antitumor activity, beta-lapachone, tanshione IIA

## Abstract

Twenty-seven L-shaped ortho-quinone analogs were designed and synthesized using a one pot double-radical synthetic strategy followed by removing methyl at C-3 of the furan ring and introducing a diverse side chain at C-2 of the furan ring. The synthetic derivatives were investigated for their cytotoxicity activities against human leukemia cells K562, prostate cancer cells PC3, and melanoma cells WM9. Compounds **TB1**, **TB3**, **TB4**, **TB6**, **TC1**, **TC3**, **TC5**, **TC9**, **TC11**, **TC12**, **TC14**, **TC15**, **TC16**, and **TC17** exhibited a better broad-spectrum cytotoxicity on three cancer cells. **TB7** and **TC7** selectively displayed potent inhibitory activities on leukemia cells K562 and prostate cancer cells PC3, respectively. Further studies indicated that **TB3**, **TC1**, **TC3**, **TC7**, and **TC17** could significantly induce the apoptosis of PC3 cells. **TC1** and **TC17** significantly induced apoptosis of K562 cells. **TC1**, **TC11**, and **TC14** induced significant apoptosis of WM9 cells. The structure-activity relationships evaluation showed that removing methyl at C-3 of the furan ring and introducing diverse side chains at C-2 of the furan ring is an effective strategy for improving the anticancer activity of L-shaped ortho-quinone analogs.

## 1. Introduction

Over several decades, cancer continues to be the most awful disease due to its uncontrolled cell growth and the fact that it is a dominate killer of human beings worldwide [1]. Especially in China, millions of deaths have been caused by tumor. The common cancer types in Chinese male, in 2018, were lung, stomach, colorectum, liver, and esophageal cancer. Additionally, breast, lung, colorectum, thyroid, and stomach cancer were the common types in Chinese female [2]. The incidence of colorectal cancer in males and females has increased, however, the incidence of esophageal, stomach, and liver cancer has decreased between 2000 and 2011 [3]. Meanwhile, the incidence and mortality of prostate cancer and bladder cancer in males, together with obesity and hormonal exposure-related cancers, namely thyroid, breast, and ovarian cancer in females have shown a rising trend [3]. However, the standardized treatments of cancer, including surgery, chemotherapy, and radiation therapy, show many limitations, such as severe adverse effects, recurrence, and increasing drug resistance [4].

Currently, phytochemicals have become a valuable source of anticancer drugs. Actually, over 75% of nonbiological anticancer drugs approved are either plant-derived natural products or developed based on these products [5]. Therefore, natural products have continued to be a hot research topic for the development of new antitumor drugs [6,7,8,9].

Tanshinone IIA is a natural ortho-quinone isolated from the rhizome of *Salvia miltiorrhiza Bunge* with antineoplastic activity, such as gastric cancer, breast cancer, osteosarcoma, etc. [10,11,12,13]. These various properties demonstrate that tanshinone IIA is a potential antitumor drug candidate. Furthermore, beta-lapachone is another natural ortho-quinone which has been reported to selectively kill many human cancer cells [14], however, the pyran ring of beta-lapachone has been found to be unstable during metabolism in the human body, and may led to side effects on normal tissues [15,16]. Recently, studies have revealed that some tanshinone analogs show similar or stronger antitumor activity when the ring-A is removed but the furan ring is retained [17,18]. You et al. [19,20] discovered that the binding site for quinone oxidoreductase-1 (NQO1) substrates was an L-shaped pocket (Figure 1B) which binds well with tanshinone analogs, and showed higher antitumor activities than the planar compound **1** and beta-lapachone. Therefore, we surmise that removing methyl at C-3 of the furan ring is more suitable for the binding site, and we anticipate that a novel L-shaped molecule without methyl at C-3 of the furan ring could provide better antitumor activities. Considering that some nitrogen, oxygen-substituted, and amino acid substrates can improve aqueous solubility and antitumor activities [21,22,23,24,25], we have attempted to introduce a great diversity of oxygen-substituted, nitrogen-containing groups and amino acids. Thus, in this work, we developed quinone-directed agents by removing methyl at C-3 of the furan ring and introducing a diverse side chain at C-2 of the furan ring, culminating in the discovery of a promising scaffold. The inhibitory activity was assessed in vitro using three cell lines including K562, PC3, and WM9.

## 2. Discussion and Results

### 2.1. Chemistry

The synthesis of two substituted naphtho[1,2-b]furan-4,5-diones is outlined in Scheme 1. Briefly, treatment of lawsone **6** with allyl bromide followed by subsequent Claisen rearrangement afforded **7**, which was then cyclized to get the ortho-quinone **8** by using Lewis acid NbCl_5_ at room temperature [26]. 

Initially, dealing **8** with *N*-bromosuccinimide (NBS) and 2,2′-azobis(2-methylpropionitrile) (AIBN) afforded only trace amounts of **8a**. Another intermediate, **8b**, was obtained by Nelson’s method [27] as shown in Scheme 1. Then, compound **9** was obtained from **8b** through a second radical reaction. Considering the same reaction condition, we successfully got **9** from **8** through a bis-radical reaction. The brominated intermediate **9** was reacted with substituted phenol or amine to provide ortho-quinone derivatives **TB1**–**TB9** and **TC1**–**TC18**, respectively. All the structures of ortho-quinone derivatives were identified through ^1^H, ^13^C, and HRMS.

In summary, we successfully established an effective synthetic strategy, which removed the methyl at C-3 of the furan ring and introduced diverse side chains at C-2 of the furan ring. In addition, we replaced the bromide of **9** with a variety of oxygen-substituted, nitrogen-containing group, and amino acid by a nucleophilic substitution.

### 2.2. In Vitro Cytotoxicity Assay

The cytotoxic activities of 5 µmol/L of the synthesized l-shaped ortho-quinone analogs were determined by using three cancer cell lines (Table 1). The results revealed that compounds **TB1**, **TB3**, **TB4**, **TB6**, **TC1**, **TC3**, **TC5**, **TC9**, **TC11**, **TC12**, **TC15**, **TC16**, and **TC17** showed a broad-spectrum potent inhibitory activity on the proliferation of the cancer cell lines, with a more than 70% inhibition rate, and **TB7** showed better inhibitory activity on K562 cells as compared with other cells. Moreover, we observed that **TC7** inhibited the growth of PC3 cells more efficiently than other cells. 

The concentration inhibition curves (Figure 2) were analyzed to calculate the IC_50_ values of the selected active compounds. The results indicated there was a dose-dependent trend of the inhibitory response of all active compounds on three cancer cells for treating 48 h. The IC_50_ values were summarized in Table 2 and show that the cytotoxicity of compounds **TB3**, **TC1**, **TC3**, **TC7**, **TC9**, and **TC17** on PC3 were better (*P* < 0.05) than that of the positive control (tanshinone IIA and paclitaxel), and another active compound exhibited similar activity to that of the positive control. The inhibition activity of **TC1** against the growth of K562 was better than that of the positive control, paclitaxel, and tanshinone IIA. Compounds **TB6**, **TC1**, **TC11**, **TC14**, and **TC15** inhibited the growth of WM9 better (*P* < 0.05) than that of tanshinone IIA and paclitaxel. In summary, most of novel L-shaped ortho-quinone analogs exhibited relatively better cytotoxicity activity as compared with the two positive controls, which indicated that the analogs containing L-shaped ortho-quinone as the core structure, possessed stronger anticancer activity. This result provided a preliminary biological activity basis for the investigation of anticancer candidate agents.

### 2.3. Structure-Activity Relationships Study

To obtain two series of analogs, we successfully built an effective synthetic strategy by removing the methyl at C-3 of the furan ring and introducing diverse side chains at C-2 of the furan ring. On the basis of the cytotoxicity results (Table 1 and Table 2), a preliminary structure-activity relationships could be established. The TB series molecules bearing electron-withdrawing groups or multi-substituted groups such as compounds **TB3**, **TB4**, and **TB6** showed a better inhibitory effect on PC3 cell lines, K562 cell lines, and WM9 cell lines, whereas the TB series molecules bearing alkane groups at the 2-position showed decreased cytotoxicity in PC3 cell lines, K562 cell lines, and WM9 cell lines, such as compounds **TB9**. The TC series molecules with electron-withdrawing groups, saturate six-membered rings, or multi-substituted groups emerged greater inhibitory effects on three cancer cell lines, such as **TC11**, **TC12**, **TC15**, and **TC16**, whereas the TC series molecules bearing donating groups or alkane groups at the 2-position showed reduced cytotoxicity in three cancer cell lines. The structure-activity relationships evaluation also showed that removing methyl at C-3 of the furan ring and introducing diverse side chains at C-2 of the furan ring were good strategies for improving the anticancer activity of L-shaped ortho-quinone analogs.

### 2.4. Effects of Active Compounds on Cell Apoptosis

According to the above IC_50_ values of all active compounds, we selected six active compounds (**TB3**, **TC1**, **TC3**, **TC7**, **TC9**, and **TC17**) for PC3, three active compounds (**TB6**, **TC1**, and **TC17**) for K562, and four active compounds (**TB6**, **TC1**, **TC11**, and **TC14**) for WM9, based on their higher activities than that of the positive control and better selectivity and, then, studied their effects on cell apoptosis by microscope observation (Figure 3) and flow cytometry (Figure 4). The microscopic observations (Figure 3A) showed that the number of PC3 cells was significantly reduced by treatments with 2.5 µmol/L of **TB3**, **TC1**, **TC3**, and **TC7**; the apoptotic bodies and cell fragments were significantly observed as compared with the control group. The PC3 cells treated with **TC9** showed that the number of cells were significantly reduced, while fewer cells died and there were no significant apoptotic bodies. The PC3 cells treated with **TC7** showed a significant decrease in the number of cells, meanwhile, some cells died, apoptotic bodies appeared obviously, and the morphology of some cells became an irregular shape of spindle length. Above all, the inhibitory activity of **TB3**, **TC1**, **TC3**, and **TC7** may be through inducing apoptosis, another two compounds may be through different types.

The K562 cells treated with **TC1** were obviously dead and dispersed, with the appearance of apoptotic bodies as comparing with the control group (Figure 3B). The cells treated with **TB6** had a significantly reduced number of cells and most cells clumped growth similar to the control cells. For the cells treated with **TC17** we observed both dead cells and fewer clumps of cells. Furthermore, for the WM9 cell lines treated with **TB6**, **TC1**, **TC11**, and **TC14** (Figure 3C), we observed that the cells treated with **TC1** and **TC11** were obviously dead with a large number of apoptotic bodies and dispersed cells; the cells treated with **TB6** showed a significant decrease in the number of cells and fewer dead cells. Observation of the cells treated with **TC14** showed that the number of cells was significantly reduced, while some cells were obviously dead with apoptotic bodies appearing, and the morphology of some cells also became an irregular shape of spindle length. The above results indicated that **TC1** can induce apoptosis for K562 and WM9 cells to inhibit the growth; **TB6**, **TC11**, **TC14**, and **TC17** can jointly inhibit the proliferation of cell through a variety of mechanisms.

Flow cytometry analyzed results (Figure 4) confirmed that **TB3** (*p* < 0.01), **TC1** (*p* < 0.01), **TC3** (*p* < 0.01), **TC7** (*p* < 0.01), and **TC17** (*p* < 0.05) could significantly induce the apoptosis of PC3 cells (Figure 4A), while **TC9** did not. **TC1** (*p* < 0.01) and **TC17** (*p* < 0.05) significantly induced apoptosis of K562 cells (Figure 4B), while **TB6** had no significant effect on apoptosis of K562 and WM9 cells. **TC1** (*p* < 0.01), **TC11** (*p* < 0.05), and **TC14** (*p* < 0.01) could potently induce apoptosis for WM9 cells (Figure 4C). 

## 3. Materials and Methods

### 3.1. Instruments and Materials

High-resolution mass spectra (HRMS) were obtained on an electrospray ionization (ESI) mode on a Bruker ESI-QTOF mass spectrometry. Nuclear magnetic resonance (NMR) spectra were recorded on a Bruker Avance NEO (^1^H NMR, 600 MHz; ^13^C NMR, 150 MHz, Bruker, Switzerland) with TMS as an internal standard. The IR spectra were recorded by using a FTIR Spectrometer (IR 200 Fourier Energy Spectrum Technology Co., Ltd., TianJin, China) and the KBr disk method was adopted. The melting points (mp) were determined on an WRX-4 microscope melting point apparatus. The column chromatography was performed on silica gel (Qingdao, 200–300 mesh) and the thin-layer (0.25 mm) chromatography (TLC) analysis was carried out on silica gel plates (Qingdao, China). Other reagents were analytical grade or guaranteed reagent commercial product and used without further purification, unless otherwise noted.

### 3.2. Methods of Synthesis

#### 3.2.1. Synthesis of 2-Allyl-3-hydroxy-1,4-naphthoquinone (**7**)

A mixture of lawsone **6** (10.0 g, 57.42 mmol) and anhydrous K_2_CO_3_ (7.94 g, 57.42 mmol) in anhydrous DMF (100 mL) were stirred for 15 min at room temperature. Allyl bromide (17.37 g, 143.55 mmol) in DMF (5 mL) was added dropwise and stirred for 15 min at 0 °C. The mixture was refluxed at 120 °C for 3 h and then cooled to room temperature before it was poured into water and extracted with EA. The organic phase was washed with brine, dried over Na_2_SO_4_, filtered, and concentrated in vacuo. The crude product was purified by column chromatography on silica gel (eluent: petroleum ether/EtOAc 15:1) to afford **7** (7.8 g, 63% yield) as a light yellow solid. Other data was found in reference [26].

#### 3.2.2. Synthesis of 2-Methyl-2,3-dihydrolnaphthol[1,2-*b*]furan-4,5-dione (**8**)

NbCl_5_ (18.92 g, 70.02 mmol) was added into **7** (3.0 g, 14.00 mmol) in anhydrous DCM (50 mL) at 0 °C. After stirring for 45 min at 30 °C, the mixture was poured into ice water and extracted with DCM. The organic phase was washed with brine, dried over Na_2_SO_4_, filtered, and concentrated in vacuo. The crude product was purified by column chromatography on silica gel (eluent: petroleum ether/EtOAc 4:1) to afford **8** (2.16 g, 72% yield) as red solid. Other data was found in reference [26].

#### 3.2.3. Synthesis of 2-Bromomethyl-naphtho[1-*b*]furan-4,5-dione (**9**)

A mixture of **8** (1.5 g, 7.00 mmol), anhydrous *N*-bromosuccinimide (2.49 g, 1.40mmol), and 2,2′-azobis(2-methylpropionitrile) (114.98 mg, 1.40 mmol) in anhydrous CCl_4_ (50 mL) was stirred under argon at 70 °C, until **8** were disappeared. Then, the mixture was cooled to room temperature, and anhydrous N-bromosuccinimide (2.49 g, 1.40 mmol) and 2,2′-azobis(2-methylpropionitrile) (114.98 mg, 1.40 mmol) were added and stirred at 70 °C for 2 h. The mixture was cooled to room temperature, and poured into water, extracted with EA, washed with brine, dried over Na_2_SO_4_, filtered, and concentrated in vacuo. The crude product was purified by column chromatography on silica gel (eluent: petroleum ether/EtOAc 12:1) to afford **9** (1.0 g, 67% yield) as red solid. Mp: 169–170 °C. ^1^H NMR (600 MHz, CDCl_3_) *δ* 8.09 (d, *J* = 7.8 Hz, 1H), 7.77 (d, *J* = 7.6 Hz, 1H), 7.69 (t, *J* = 7.6 Hz, 1H), 7.51 (t, *J* = 7.6 Hz, 1H), and 6.83 (s, 1H), 4.55 (s, 2H). ^13^C NMR (150 MHz, CDCl_3_) *δ* 180.06, 174.13, 160.94, 153.46, 135.56, 130.74, 130.69, 129.03, 127.98, 122.69, 122.34, 108.04, and 21.87. IR (ν, cm^−1^): 3118.33, 2920.28, 2339.32, 1670.84, 1551.16, and 691.42. HRMS (ESI) calcd. for [M + Na]^+^ C_13_H_7_O_3_BrNa^+^: 312.9471, found 312.9460.

#### 3.2.4. Synthesis of **TB1**–**9** and **TC1**–**18**

A mixture of the corresponding amines or alcohols (0.52 mmol), K_2_CO_3_ (142 mg, 1.03 mmol), and **9** (100 mg, 0.34 mmol) in THF (5 mL) was stirred at 30 °C to 50 °C for 4 h. After cooling, the mixture was poured into water and extracted with EA. The combined organic layer was washed with brine and dried over anhydrous Na_2_SO_4_, filtered, and concentrated to afford a crude product which was purified through column chromatography on silica gel.

**TB1**: *2-(((4-methoxyphenyl)oxy)methyl)naphtho[1-b]furan-4,5-dione.* Red solid, yield: 34%. Mp: 112–113 °C. ^1^H NMR (600 MHz, CDCl_3_) *δ* 8.09 (d, *J* = 7.6 Hz, 1H), 7.75 (d, *J* = 7.6 Hz, 1H), 7.66 (t, *J* = 7.5 Hz, 1H), 7.48 (t, *J* = 7.6 Hz, 1H), 6.94–6.92 (m, 2H), 6.87–6.85 (m, 2H), 6.83 (s, 1H), 5.04 (s, 2H), and 3.78 (s, 3H). ^13^C NMR (150 MHz, CDCl_3_) *δ* 180.34, 174.38, 160.88, 154.72, 153.82, 151.95, 135.47, 130.66, 130.50, 128.98, 128.28, 122.59, 122.12, 116.34, 114.81, 108.37, 63.03, and 55.73. IR (ν, cm^−1^): 3445.32, 2358.40, 2339.32, 1671.47, 1598.48, 1253.20, 1219.81, 1161.58, and 1057.34. HRMS (ESI) calcd. for [M + Na]^+^ C_20_H_14_O_5_Na^+^: 357.0733, found 357.0723.

**TB2**: *2-(((4-acetylphenyl)oxy)methyl)naphtho[1,2-b]furan-4,5-dione.* Light orange solid, yield: 40%. Mp: 190–191 °C. ^1^H NMR (600 MHz, CDCl_3_) *δ* 8.13 (d, *J* = 6.4 Hz, 1H), 8.00 (d, *J* = 8.9 Hz, 2H), 7.78 (d, *J* = 9.1 Hz, 1H), 7.69 (t, *J* = 7.6 Hz, 1H), 7.53 (t, *J* = 7.6 Hz, 1H), 7.06 (d, *J* = 8.9 Hz, 2H), 6.94 (s, 1H), 5.19 (s, 2H), and 2.60 (s, 3H). ^13^C NMR (150 MHz, CDCl_3_) *δ* 196.71, 180.20, 174.31, 161.58, 161.10, 152.60, 135.53, 131.28, 130.74, 130.70, 129.02, 128.11, 122.64, 122.08, 114.48, 108.98, 61.92, and 26.43. IR (ν, cm^−1^): 3445.93, 2358.51, 2341.16, 1671.86, 1599.84, 1249.41, 1217.01, 1178.57, and 1008.47. HRMS (ESI) calcd. for [M + Na]^+^ C_21_H_14_O_5_Na^+^: 369.0733, found 369.0721.

**TB3**: *2-(((4-propiophenyl)oxy)methyl)naphtho[1,2-b]furan-4,5-dione.* Orange solid, yield: 37%. Mp: 179–180 °C. ^1^H NMR (600 MHz, CDCl_3_) *δ* 8.11 (d, *J* = 7.6 Hz, 1H), 8.00 (d, *J* = 8.9 Hz, 2H), 7.77 (d, *J* = 6.4 Hz, 1H), 7.71–7.68 (m, 1H), 7.53–7.50 (m, 1H), 7.06 (d, *J* = 8.9 Hz, 2H), 6.93 (s, 1H), 5.18 (s, 2H), 2.99 (q, *J* = 7.2 Hz, 2H), and 1.24 (t, *J* = 7.3 Hz, 3H). ^13^C NMR (150 MHz, CDCl_3_) *δ* 199.42, 180.20, 174.30, 161.40, 161.06, 152.68, 135.53, 130.98, 130.72, 130.68, 130.36, 129.01, 128.11, 122.63, 122.08, 114.47, 108.92, 61.90, 31.53, and 8.38. IR (ν, cm^−1^): 3447.03, 2358.62, 2341.16, 1669.65, 1601.07, 1222.31, 1180.35, and 1002.86. HRMS (ESI) calcd. for [M + Na]^+^ C_22_H_16_O_5_Na^+^: 383.0890, found 383.0876.

**TB4**: *2-(((4-nitrophenyl)oxy)methyl)naphtho[1,2-b]furan-4,5-dione.* Red solid, yield: 48%. Mp: 205–206 °C. ^1^H NMR (600 MHz, DMSO-*d_6_*) *δ* 8.25 (d, *J* = 9.3 Hz, 2H), 7.96 (d, *J* = 7.6 Hz, 1H), 7.77–7.74 (m, 2H), 7.59–7.55 (m, 1H), 7.34 (d, *J* = 9.3 Hz, 2H), 7.18 (s, 1H), and 5.42 (s, 2H). ^13^C NMR (150 MHz, DMSO-*d_6_*) *δ* 179.58, 174.56, 163.29, 160.07, 152.42, 141.83, 135.43, 130.84, 130.17, 129.83, 127.89, 126.41, 122.50, 122.36, 115.94, 110.32, and 62.45. IR (ν, cm^−1^): 3445.75, 2358.70, 2341.16, 1681.37,1592.17, 1507.95, 1384.15, 1340.19, 1277.67, and 1110.13. HRMS (ESI) calcd. for [M + Na]^+^ C_19_H_11_O_6_NNa^+^: 372.0479, found 372.0465.

**TB5**: *2-(((2-methoxyl-4-formyl)phenyl)oxy)methyl)naphtho[1,2-b]furan-4,5-dione.* Red solid, yield: 34%. Mp: 203–204 °C. ^1^H NMR (600 MHz, CDCl_3_) *δ* 9.91 (s, 1H), 8.13 (d, *J* = 6.4 Hz, 1H), 7.77 (d, *J* = 7.6 Hz, 1H), 7.72–7.68 (m, 1H), 7.54–7.48 (m, 3H), 7.15 (d, *J* = 8.0 Hz, 1H), 6.96 (s, 1H), 5.28 (s, 2H), and 3.97 (s, 3H). ^13^C NMR (150 MHz, CDCl_3_) *δ* 190.88, 180.19, 174.31, 161.15, 152.50, 152.42, 150.30, 135.52, 131.28, 130.75, 129.67, 129.03, 128.10, 122.68, 122.09, 115.30, 112.93, 109.87, 109.40, 62.85, and 56.09. IR (ν, cm^−1^): 3446.32, 2358.18, 2341.16, 1702.84, 1676.79, 1586.53, 1508.23, 1267.05, 1236.25, 1137.10, and 999.76. HRMS (ESI) calcd. for [M + Na]^+^ C_21_H_14_O_6_Na^+^: 385.0683, found 385.0668.

**TB6**: *2-(((4-formylphenyl)oxy)methyl)naphtho[1,2-b]furan-4,5-dione.* Orange solid, yield: 30%. Mp: 212–213 °C. ^1^H NMR (600 MHz, CDCl_3_) *δ* 9.95 (s, 1H), 8.13 (d, *J* = 7.8 Hz, 1H), 7.92 (d, *J* = 8.9 Hz, 2H), 7.78 (d, *J* = 7.8 Hz, 1H), 7.73–7.69 (m, 1H), 7.56–7.51 (m, 1H), 7.14 (d, *J* = 8.7 Hz, 2H), 6.96 (s, 1H), and 5.21 (s, 2H). ^13^C NMR (150 MHz, CDCl_3_) *δ* 190.71, 180.17, 174.30, 162.67, 161.14, 152.36, 135.53, 132.10, 130.82, 130.77, 130.74, 129.05, 128.09, 122.64, 122.08, 115.06, 109.11, and 61.99. IR (ν, cm^−1^): 3446.21, 2358.24, 2337.30, 1687.41, 1671.45, 1598.48, 1253.19, 1219.95, 1161.49, and 1057.40. HRMS (ESI) calcd. for [M + Na]^+^ C_20_H_12_O_5_Na^+^: 355.0577, found 355.0565.

**TB7**: *2-(((4-bromo-2-formyl)phenyl)oxy)methyl)naphtho[1,2-b]furan-4,5-dione.* Light orange solid, yield: 33%. Mp: 201–202 °C. ^1^H NMR (600 MHz, CDCl_3_) *δ* 10.43 (s, 1H), 8.14 (d, *J* = 7.6 Hz, 1H), 7.99 (d, *J* = 2.7 Hz, 1H), 7.76 (d, *J* = 7.6 Hz, 1H), 7.73–7.69 (m, 2H), 7.57–7.51 (m, 1H), 7.06 (d, *J* = 8.7 Hz, 1H), 6.96 (s, 1H), and 5.24 (s, 2H). ^13^C NMR (150 MHz, CDCl_3_) *δ* 187.86, 180.06, 174.25, 161.30, 158.94, 151.80, 138.27, 135.59, 131.52, 130.87, 130.82, 129.07, 127.95, 126.84, 122.64, 121.99, 115.01, 114.95, 109.43, and 62.80. IR (ν, cm^−1^): 3445.63, 2358.34, 2337.30, 1671.90, 1599.56, 1556.29, 1249.42, 1217.50, 1178.60, and 1008.51. HRMS (ESI) calcd. for [M + Na]^+^ C_20_H_11_O_5_BrNa^+^: 432.9682, found 432.9673.

**TB8**: *2-(methoxymethyl)naphtho[1,2-b]furan-4,5-dione.* Red solid, yield: 12%. Mp: 53–54 °C. ^1^H NMR (600 MHz, CDCl_3_) *δ* 8.11 (dd, *J* = 7.8, 1.5 Hz, 1H), 7.78 (dd, *J* = 7.7, 1.4 Hz, 1H), 7.70–7.66 (m, 1H), 7.52–7.48 (m, 1H), 6.80 (s, 1H), 4.51 (s, 2H), and 3.46 (s, 3H). ^13^C NMR (150 MHz, CDCl_3_) *δ* 180.43, 174.46, 160.91, 154.77, 135.45, 130.64, 130.44, 128.96, 128.34, 122.59, 122.05, 107.97, 66.02, and 58.32. IR (ν, cm^−1^): 3446.10, 2358.55, 2337.96, 1677.60, 1276.65, 1216.21, 1153.22, and 1082.89. HRMS (ESI) calcd. for [M + Na]^+^ C_14_H_10_O_4_Na^+^: 265.0471, found 265.0463.

**TB9**: *2-(ethoxymethyl)naphtho[1,2-b]furan-4,5-dione.* Red solid, yield: 11%. Mp: 60–61 °C. ^1^H NMR (600 MHz, CDCl_3_) *δ* 8.10 (dd, *J* = 8.2, 1.3 Hz, 1H), 7.77 (dd, *J* = 7.6, 1.2 Hz, 1H), 7.71–7.65 (m, 1H), 7.52–7.46 (m, 1H), 6.78 (s, 1H), 4.55 (s, 2H), 3.63 (q, *J* = 7.0 Hz, 2H), and 1.29 (t, *J* = 7.0 Hz, 3H). ^13^C NMR (150 MHz, CDCl_3_) *δ* 180.47, 174.45, 160.81, 155.23, 135.45, 130.61, 130.37, 128.90, 128.39, 122.59, 122.08, 107.66, 66.26, 64.25, and 15.10. IR (ν, cm^−1^): 3446.08, 2358.74, 2342.78, 1681.54, 1275.43, 1215.49, 1159.01, and 1083.47. HRMS (ESI) calcd. for [M + Na]^+^ C_15_H_12_O_4_Na^+^: 279.0628, found 279.0621.

**TC1**: *2-(diethylaminomethyl)naphtho[1,2-b]furan-4,5-dione.* Red solid, yield: 51%. Mp: 78–79 °C. ^1^H NMR (600 MHz, CDCl_3_) *δ* 8.07 (d, *J* = 7.6 Hz, 1H), 7.72 (d, *J* = 7.6 Hz, 1H), 7.65 (t, *J* = 7.0 Hz, 1H), 7.46 (t, *J* = 7.6 Hz, 1H), 6.66 (s, 1H), 3.77 (s, 2H), 2.62 (q, *J* = 7.2 Hz, 4H), and 1.14 (t, *J* = 7.2 Hz, 6H). ^13^C NMR (150 MHz, CDCl_3_) *δ* 180.65, 174.55, 160.34, 156.44, 135.41, 130.55, 130.12, 128.74, 128.58, 122.45, 122.21, 107.11, 48.69, 47.10, and 12.01. IR (ν, cm^−1^): 3445.24, 2953.88, 2358.48, 2339.23, 1700.91, 1676.52, 1216.10, and 1111.39. HRMS (ESI) calcd. for [M + Na]^+^ C_17_H_18_O_3_Na^+^: 284.1281, found 284.1271.

**TC2**: *2-(diisopropylaminomethyl)naphtho[1,2-b]furan-4,5-dione*. Red solid, yield: 18%. Mp: 69–70 °C. ^1^H NMR (600 MHz, CDCl_3_) *δ* 8.06 (d, *J* = 7.8 Hz, 1H), 7.68 (d, *J* = 6.2 Hz, 1H), 7.64 (t, *J* = 7.5 Hz, 1H), 7.44 (t, *J* = 6.7 Hz, 1H), 6.66 (s, 1H), 3.72 (s, 2H), 3.13 (p, *J* = 6.5 Hz, 2H), and 1.08 (d, *J* = 6.7 Hz, 12H). ^13^C NMR (150 MHz, CDCl_3_) *δ* 180.88, 174.66, 161.17, 159.76, 135.37, 130.50, 129.81, 128.91, 128.65, 122.50, 122.09, 105.12, 49.12, 42.36, and 20.81. IR (ν, cm^−1^): 3445.80, 2966.49, 2358.61, 2337.30, 1676.07, 1558.32, 1215.85, and 1149.51. HRMS (ESI) calcd. for [M + Na]^+^ C_19_H_22_O_3_Na^+^: 312.1594, found 312.1583.

**TC3**: *2-((l-methionine methyl ester-1-yl)methyl)naphtho[1,2-b]furan-4,5-dione.* Red solid, yield: 12%. Mp: 52–53 °C. ^1^H NMR (600 MHz, CDCl_3_) *δ* 8.08 (d, *J* = 7.8 Hz, 1H), 7.72 (d, *J* = 7.6 Hz, 1H), 7.66 (t, *J* = 7.6 Hz, 1H), 7.47 (t, *J* = 7.6 Hz, 1H), 6.66 (s, 1H), 3.95 (d, *J* = 15.1 Hz, 1H), 3.79 (d, *J* = 15.1 Hz, 1H), 3.74 (s, 3H), 3.52 (dd, *J* = 8.4, 5.0 Hz, 1H), 2.64 (t, *J* = 7.2 Hz, 2H), 2.10 (s, 3H), 2.02–1.97 (m, 1H) and 1.89–1.82 (m, 1H). ^13^C NMR (150 MHz, CDCl_3_) *δ* 180.55, 175.18, 174.44, 160.36, 157.00, 135.44, 130.58, 130.19, 128.80, 128.47, 122.35, 122.17, 106.00, 59.14, 52.14, 44.65, 32.68, 30.48, and 15.38. IR (ν, cm^−1^): 3446.14, 2923.56, 2358.62, 2335.37, 1698.61, 1670.40, 1215.45, and 1147.65. HRMS (ESI) calcd. for [M + Na]^+^ C_19_H_19_O_5_NSNa^+^: 396.0876, found 396.0865.

**TC4**: *2-((l-alanine methyl ester-1-yl)methyl)naphtho[1,2-b]furan-4,5-dione.* Red solid, yield: 27%. Mp: 83–84 °C. ^1^H NMR (600 MHz, CDCl_3_) *δ* 8.07 (d, *J* = 6.9 Hz, 1H), 7.73 (d, *J* = 4.9 Hz, 1H), 7.65 (d, *J* = 8.6 Hz, 1H), 7.46 (t, *J* = 6.6 Hz, 1H), 6.66 (s, 1H), 3.94 (d, *J* = 14.1 Hz, 1H), 3.81 (d, *J* = 14.9 Hz, 1H), 3.74 (s, 3H), 3.46 (q, *J* = 6.9 Hz, 1H), and 1.37 (d, *J* = 6.6 Hz, 3H). ^13^C NMR (150 MHz, CDCl_3_) *δ* 180.52, 175.63, 174.42, 160.36, 156.93, 135.40, 130.56, 130.18, 128.80, 128.46, 122.39, 122.18, 105.99, 55.69, 52.05, 44.26, and 19.13. IR (ν, cm^−1^): 3328.35, 2958.60, 2358.44, 2337.30, 1735.12, 1676.39, 1216.30, and 1139.72. HRMS (ESI) calcd. for [M + Na]^+^ C_17_H_15_O_5_NNa^+^: 336.0842, found 336.0831.

**TC5**: *2-((l-isoleucinate methyl ester-1-yl)methyl)naphtho[1,2-b]furan-4,5-dione.* Red solid, yield: 39%. Mp: 112–113 °C. ^1^H NMR (600 MHz, CDCl_3_) *δ* 8.07 (d, *J* = 7.7 Hz, 1H), 7.71 (d, *J* = 7.7 Hz, 1H), 7.67–7.62 (m, 1H), 7.48–7.43 (m, 1H), 6.65 (s, 1H), 3.92 (d, *J* = 15.1 Hz, 1H), 3.72 (d, *J* = 16.4 Hz, 4H), 3.17 (d, *J* = 5.8 Hz, 1H), 1.57–1.50 (m, 1H), 1.26–1.16 (m, 2H), and 0.93–0.88 (m, 6H). ^13^C NMR (150 MHz, CDCl_3_) *δ* 180.60, 175.16, 174.46, 160.27, 157.37, 135.45, 130.55, 130.15, 128.78, 128.52, 122.32, 122.19, 105.82, 65.31, 51.64, 45.08, 38.47, 25.39, 15.66, and 11.49. IR (ν, cm^−1^): 3445.72, 2924.70, 2358.43, 2341.16,1732.42, 1682.86, 1209.15, and 1150.10. HRMS (ESI) calcd. for [M + Na]^+^ C_20_H_21_O_5_NNa^+^: 378.1312, found 378.1299.

**TC6**: *2-((l-valine methyl ester-1-yl)methyl)naphtho[1,2-b]furan-4,5-dione*. Red solid, yield: 41%. Mp: 54–55 °C. ^1^H NMR (600 MHz, CDCl_3_) *δ* 8.09 (d, *J* = 6.2 Hz, 1H), 7.73 (d, *J* = 7.6 Hz, 1H), 7.70–7.64 (m, 1H), 7.50–7.45 (m, 1H), 6.67 (s, 1H), 3.95 (d, *J* = 15.3 Hz, 1H), 3.77–3.72 (m, 4H), 3.11 (d, *J* = 5.8 Hz, 1H), 2.02–1.95 (m, 1H), and 0.98 (t, *J* = 6.3 Hz, 7H). ^13^C NMR (150 MHz, CDCl_3_) *δ* 180.62, 175.22, 174.49, 160.28, 157.39, 135.43, 130.59, 130.14, 128.82, 128.56, 122.32, 122.22, 105.83, 66.36, 51.70, 45.16, 31.75, 19.28, and 18.35. IR (ν, cm^−1^): 3425.42, 2923,62, 2358.56, 2337.30, 1698.57, 1670.37, 1187.10, and 1118.00. HRMS (ESI) calcd. for [M + Na]^+^ C_19_H_19_O_5_NNa^+^: 364.1155, found 364.1141.

**TC7**: *2-((l-glycine methyl ester-1-yl)methyl)naphtho[1,2-b]furan-4,5-dione*. Red solid, yield: 17%. Mp: 55–56 °C. ^1^H NMR (600 MHz, CDCl_3_) *δ* 8.06 (d, *J* = 7.6 Hz, 1H), 7.71 (d, *J* = 6.4 Hz, 1H), 7.68–7.62 (m, 1H), 7.49–7.43 (m, 1H), 6.67 (s, 1H), 3.93 (s, 2H), 3.75 (s, 3H), and 3.50 (s, 2H). ^13^C NMR (150 MHz, CDCl_3_) *δ* 180.49, 174.42, 172.45, 160.49, 156.62, 135.44, 130.57, 130.25, 128.79, 128.39, 122.41, 122.14, 106.25, 52.03, 49.50, and 45.38. IR (ν, cm^−1^): 3328.36, 2958.26, 2358.57, 2337.30, 1735.10, 1676.24, 1216.29, and 1180.25. HRMS (ESI) calcd. for [M + Na]^+^ C_16_H_13_O_5_NNa^+^: 322.0686, found 322.0680.

**TC8**: *2-((l-leucinate methyl ester-1-yl)methyl)naphtho[1,2-b]furan-4,5-dione*. Red solid, yield: 19%. Mp: 54–55 °C. ^1^H NMR (600 MHz, CDCl_3_) *δ* 8.09 (d, *J* = 7.6 Hz, 1H), 7.73 (d, *J* = 7.6 Hz, 1H), 7.67 (t, *J* = 7.6 Hz, 1H), 7.47 (t, *J* = 7.6 Hz, 1H), 6.66 (s, 1H), 3.94 (d, *J* = 15.1 Hz, 1H), 3.77 (d, *J* = 15.1 Hz, 1H), 3.73 (s, 3H), 3.38 (t, *J* = 7.2 Hz, 1H), 1.82–1.77 (m, 1H), 1.52 (t, *J* = 7.4 Hz, 2H), 0.95 (d, *J* = 6.7 Hz, 3H), and 0.90 (d, *J* = 6.7 Hz, 3H). ^13^C NMR (150 MHz, CDCl_3_) *δ* 180.59, 176.01, 174.47, 160.32, 157.18, 135.42, 130.59, 130.17, 128.84, 128.51, 122.32, 122.20, 105.94, 59.08, 51.88, 44.63, 42.75, 24.89, 22.78, and 22.09. IR (ν, cm^−1^): 3434.68, 2958.49, 2358.68, 2339.23, 1670.06, 1518.72, 1211.01, and 1107.62. HRMS (ESI) calcd. for [M + Na]^+^ C_20_H_21_O_5_NNa^+^: 378.1312, found 378.1302.

**TC9**: *2-((4-boc-piperazin-1-yl)methyl)naphtho[1,2-b]furan-4,5-dione*. Red solid, yield: 26%. Mp: 56–57 °C. ^1^H NMR (600 MHz, CDCl_3_) *δ* 8.10 (d, *J* = 7.4 Hz, 1H), 7.76 (d, *J* = 7.6 Hz, 1H), 7.67 (t, *J* = 7.6 Hz, 1H), 7.48 (t, *J* = 7.6 Hz, 1H), 6.70 (s, 1H), 3.68 (s, 2H), 3.49 (s, 4H), 2.52 (s, 4H), and 1.47 (s, 9H). ^13^C NMR (150 MHz, CDCl_3_) *δ* 180.53, 174.48, 160.57, 155.01, 154.68, 135.44, 130.64, 130.29, 128.81, 128.46, 122.53, 122.17, 107.66, 79.88, 54.60, 52.60, and 28.42. IR (ν, cm^−1^): 3434.73, 2358.58, 2339.23, 1669.96, 1518.77, 1211.18, and 1107.93. HRMS (ESI) calcd. for [M + H]^+^ C_22_H_25_O_5_N_2_^+^: 397.1758, found 397.1751.

**TC10**: *2-((pyrrolidin-1-yl)methyl)naphtho[1,2-b]furan-4,5-dione*. Red solid, yield: 24%. Mp: 62–63 °C. ^1^H NMR (600 MHz, CDCl_3_) *δ* 8.08 (d, *J* = 7.8 Hz, 1H), 7.76 (d, *J* = 7.6 Hz, 1H), 7.65 (t, *J* = 7.6 Hz, 1H), 7.46 (t, *J* = 7.6 Hz, 1H), 6.68 (s, 1H), 3.76 (s, 2H), 2.69–2.63 (m, 4H), and 1.89–1.83 (m, 4H). ^13^C NMR (150 MHz, CDCl_3_) *δ* 180.64, 174.54, 160.39, 156.44, 135.40, 130.56, 130.14, 128.80, 128.56, 122.55, 122.24, 106.71, 54.01, 51.84, and 23.53. IR (ν, cm^−1^): 3388.30, 3110.62, 2358.68, 2337.30, 1660.65, 1510.49, 1222.52, and 1091.51. HRMS (ESI) calcd. for [M + H]^+^ C_17_H_16_O_3_N^+^: 282.1125, found 282.1115.

**TC11**: *2-(morpholinomethyl)naphtho[1,2-b]furan-4,5-dione*. Red solid, yield: 38%. Mp: 98–99 °C. ^1^H NMR (600 MHz, CDCl_3_) *δ* 8.09 (d, *J* = 7.6 Hz, 1H), 7.76 (d, *J* = 7.6 Hz, 1H), 7.69–7.64 (m, 1H), 7.51–7.45 (m, 1H), 6.70 (s, 1H), 3.78–3.74 (m, 4H), 3.66 (s, 2H), and 2.61–2.55 (m, 4H). ^13^C NMR (150 MHz, CDCl_3_) *δ* 180.50, 174.46, 160.50, 155.01, 135.36, 130.59, 130.23, 128.86, 128.47, 122.46, 122.21, 107.64, 66.79, 54.93, and 53.28. IR (ν, cm^−1^): 3438.84, 2807.27, 2358.54, 2342.76, 1676.47, 1557.77, 1215.99, 1111.32, and 1006.63. HRMS (ESI) calcd. for [M + Na]^+^ C_17_H_15_O_4_NNa^+^: 320.0893, found 320.0886.

**TC12**: *2-(((4-fluorophenyl)amino)methyl)naphtho[1,2-b]furan-4,5-dione*. Dark red solid, yield: 51%. Mp: 174–175 °C. ^1^H NMR (600 MHz, DMSO-*d_6_*) *δ* 7.92 (d, *J* = 7.3 Hz, 1H), 7.73 (t, *J* = 7.5 Hz, 1H), 7.65 (d, *J* = 7.6 Hz, 1H), 7.52 (t, *J* = 7.5 Hz, 1H), 6.94 (t, *J* = 8.9 Hz, 2H), 6.74 (s, 1H), 6.73–6.68 (m, 2H), 6.20 (t, *J* = 6.4 Hz, 1H), and 4.37 (d, *J* = 6.0 Hz, 2H). ^13^C NMR (150 MHz, DMSO-*d_6_*) *δ* 179.86, 174.62, 159.03, 157.32, 155.16 (d, *J* = 231.0 Hz), 145.06, 135.47, 130.37, 129.85, 129.69, 128.27, 122.42, 122.06, 115.77 (d, *J* = 21.0 Hz), 113.88 (d, *J* = 6.0 Hz), and 105.93. IR (ν, cm^−1^): 3378.05, 2923.56, 2358.52, 2335.37, 1662.70, 1514.18, 1215.45, and 1161.45. HRMS (ESI) calcd. for [M + Na]^+^ C_19_H_12_O_3_NFNa^+^: 344.0693, found 344.0681.

**TC13**: *2-(((3-fluorophenyl)amino)methyl)naphtho[1,2-b]furan-4,5-dione*. Dark red solid, yield: 45%. Mp: 169–170 °C. ^1^H NMR (600 MHz, DMSO-*d_6_*) *δ* 7.92 (d, *J* = 7.6 Hz, 1H), 7.74 (t, *J* = 7.5 Hz, 1H), 7.65 (d, *J* = 7.6 Hz, 1H), 7.53 (t, *J* = 7.6 Hz, 1H), 7.10 (q, *J* = 8.0 Hz, 1H), 6.78 (s, 1H), 6.61 (t, *J* = 6.3 Hz, 1H), 6.54 (d, *J* = 8.2 Hz, 1H), 6.50 (d, *J* = 12.2 Hz, 1H), 6.35 (t, *J* = 8.4 Hz, 1H), and 4.41 (d, *J* = 6.2 Hz, 2H). ^13^C NMR (150 MHz, DMSO-*d_6_*) *δ* 179.83, 174.63, 163.90 (d, *J* = 238.5 Hz), 159.09, 156.86, 150.51 (d, *J* = 10.5 Hz), 135.46, 130.79 (d, *J* = 9.0 Hz), 130.40, 129.85, 129.73, 128.25, 122.42, 122.04, 109.34, 106.10, 102.99 (d, *J* = 21.0 Hz), and 99.24 (d, *J* = 27.0 Hz). IR (ν, cm^−1^): 3390.68, 3105.83, 2360.44, 2337.30, 1665.28, 1618.38, 1220.09, and 1152.73. HRMS (ESI) calcd. for [M + Na]^+^ C_19_H_12_O_3_NFNa^+^: 344.0693, found 344.0688.

**TC14**: *2-(((2-fluorophenyl)amino)methyl)naphtho[1,2-b]furan-4,5-dione*. Dark red solid, yield: 46%. Mp: 170–171 °C. ^1^H NMR (600 MHz, CDCl_3_) *δ* 8.08 (d, *J* = 7.8 Hz, 1H), 7.69 (d, *J* = 7.4 Hz, 1H), 7.66 (t, *J* = 7.4 Hz, 1H), 7.47 (t, *J* = 7.5 Hz, 1H), 7.05–7.00 (m, 2H), 6.79 (t, *J* = 8.5 Hz, 1H), 6.75–6.72 (m, 1H), 6.71 (s, 1H), and 4.51 (d, *J* = 6.4 Hz, 2H). ^13^C NMR (150 MHz, CDCl_3_) *δ* 180.39, 174.38, 160.36, 155.98, 151.74 (d, *J* = 237.0 Hz), 135.45, 135.22 (d, *J* = 12.0 Hz), 130.63, 130.29, 128.79, 128.36, 124.67 (d, *J* = 4.5 Hz), 122.26 (d, *J* = 15.0 Hz), 118.16 (d, *J* = 6.0 Hz), 114 (d, *J* = 19.5 Hz), 112.52, 112.50, 106.03, and 40.74. IR (ν, cm^−1^): 3425.04, 2923.94, 2358.97, 2854,13, 1670.34, 1513.50, 1186.96, and 1117.57. HRMS (ESI) calcd. for [M + Na]^+^ C_19_H_12_O_3_NFNa^+^: 344.0693, found 344.0684.

**TC15**: *2-(((2,4-difluorophenyl)amino)methyl)naphtho[1,2-b]furan-4,5-dione*. Red solid, yield: 33%. Mp: 154–155 °C. ^1^H NMR (600 MHz, CDCl_3_) *δ* 8.10 (d, *J* = 7.6 Hz, 1H), 7.71 (d, *J* = 7.6 Hz, 1H), 7.67 (t, *J* = 7.4 Hz, 1H), 7.49 (t, *J* = 7.4 Hz, 1H), 7.14 (t, *J* = 8.1 Hz, 1H), 6.77 (d, *J* = 8.0 Hz, 1H), 6.71 (s, 2H), 6.59 (d, *J* = 8.2 Hz, 1H), and 4.46 (d, *J* = 6.4 Hz, 2H). ^13^C NMR (150 MHz, CDCl_3_) *δ* 180.36, 174.38, 160.42, 155.83 (d, *J* = 9.0 Hz), 154.24 (d, *J* = 10.5 Hz), 151.95 (d, *J* = 12.0 Hz), 150.34 (d, *J* = 12.0 Hz), 135.49, 131.73 (d, *J* = 12.0 Hz), 130.52 (d, *J* = 46.5 Hz), 129.67, 128.55 (d, *J* = 73.5 Hz), 122.24 (d, *J* = 24 Hz), 115.33, 112.73 (dd, *J* = 12.0 Hz), 110.86 (dd, *J* = 25.5 Hz), 106.09, 103.90 (dd, *J* = 49.5 Hz), and 41.22. IR (*ν*, cm^−1^): 3434.72, 2358.46, 2337.30, 1689.86, 1518.53, 1210.93, 1107.68, and 957.02. HRMS (ESI) calcd. for [M + Na]^+^ C_19_H_11_O_3_NF_2_Na^+^: 362.0599, found 362.0589.

**TC16**: *2-(((2,4,6-trimethylphenyl)amino)methyl)naphtho[1,2-b]furan-4,5-dione.* Red solid, yield: 33%. Mp: 173–174 °C. ^1^H NMR (600 MHz, CDCl_3_) *δ* 8.10 (d, *J* = 7.6 Hz, 1H), 7.67 (d, *J* = 3.8 Hz, 2H), 7.50–7.46 (m, 1H), 6.85 (s, 2H), 6.60 (s, 1H), 4.21 (s, 2H), 2.29 (s, 6H), and 2.25 (s, 3H). ^13^C NMR (150 MHz, CDCl_3_) *δ* 180.53, 174.47, 160.18, 157.50, 141.61, 135.47, 132.42, 130.65, 130.23, 130.13, 129.64, 128.84, 128.47, 122.23, 122.18, 105.72, 44.94, 20.62, and 18.23. IR (*ν*, cm^−1^): 3445.76, 2357.38, 2327.66, 1670.28, 1557.44, 1215.32, 1147.47, and 1025.94. HRMS (ESI) calcd. for [M + Na]^+^ C_22_H_19_O_3_NNa^+^: 368.1257, found 368.1250.

**TC17**: *2-(((4-chlorophenyl)amino)methyl)naphtho[1,2-b]furan-4,5-dione.* Dark red solid, yield: 28%. Mp: 205–206 °C. ^1^H NMR (600 MHz, DMSO-*d_6_*) *δ* 7.92 (d, *J* = 6.8 Hz, 1H), 7.74 (t, *J* = 7.5 Hz, 1H), 7.65 (d, *J* = 7.6 Hz, 1H), 7.52 (t, *J* = 7.1 Hz, 1H), 7.12 (d, *J* = 8.9 Hz, 2H), 6.75 (s, 1H), 6.72 (d, *J* = 8.9 Hz, 2H), 6.49 (t, *J* = 6.3 Hz, 1H), and 4.40 (d, *J* = 6.2 Hz, 2H). ^13^C NMR (150 MHz, DMSO-*d*_6_) *δ*: 179.82, 174.61, 159.06, 156.95, 147.33, 135.46, 130.39, 129.84, 129.72, 129.09, 128.24, 122.41, 122.05, 120.26, 114.42, and 106.08. IR (ν, cm^−1^): 3388.22, 3110.62, 2358.84, 2342.68, 1660.69, 1510.19, 1222.68 and 1118.65. HRMS (ESI) calcd. for [M + Na]^+^ C_19_H_12_O_3_NClNa^+^, 360.0398, found 360.0384. 

**TC18**: *2-(((4-methoxyphenyl)amino)methyl)naphtho[1,2-b]furan-4,5-dione*. Dark solid, yield: 64%. Mp: 112–113 °C. ^1^H NMR (600 MHz, CDCl_3_) *δ* 8.08 (d, *J* = 6.4 Hz, 1H), 7.70 (d, *J* = 7.6 Hz, 1H), 7.69–7.63 (m, 1H), 7.50–7.44 (m, 1H), 6.82 (d, *J* = 8.9 Hz, 2H), 6.70–6.67 (m, 3H), 4.42 (s, 2H), and 3.77 (s, 3H). ^13^C NMR (150 MHz, CDCl_3_) *δ* 180.49, 174.43, 160.22, 156.83, 152.98, 140.82, 135.43, 130.62, 130.21, 128.77, 128.46, 122.27, 115.00, 114.74, 105.84, 55.77, and 42.13. IR (ν, cm^−1^): 3370.45, 3105.23, 2358.89, 2337.30, 1660.16, 1514.41, 1234.80, and 1040.25. HRMS (ESI) calcd. for [M + Na]^+^ C_20_H_15_O_4_NNa^+^: 356.0893, found 356.0883.

### 3.3. In Vitro Cytotoxicity Assay

The *human cancer cell lines*, including prostate cancer cells PC3, leukemia cells K562, and melanoma cells WM9, were stored in the biology laboratory of the Key Laboratory of Chemistry for Natural Products of Guizhou Province and Chinese Academy of Sciences (Guiyang, China). All cells were cultured in DMEM supplemented with 10% fetal bovine serum (FBS) and 1% penicillin and streptomycin (Sijiqing, Hangzhou, China) and incubated at 37 °C under 5% CO_2_, 95% air, and 95% humidity. Cytotoxicity was evaluated by performing the MTT assay [24]. Briefly, the cells were seeded in 96-well microculture plates at a density from 4 × 10^3^ to 8 × 10^3^ cells/well. Cells were then exposed to different concentrations of the assayed compounds for 48 h. Then, 20 µL of MTT solution (5 mg/mL) was added to each well and incubated at 37 °C for an additional 4 h. The medium was then removed and 200 µL Tris-DMSO solution was added. Plates were lightly shaking up to dissolve the dark blue formazan crystals and the absorbance was measured in an ELISA plate reader at 570 nm. 

### 3.4. Flow Cytometry Assay

Cell apoptosis was determined by an inverted fluorescence microscope observation and flow cytometry as describe in our previous study [28]. Briefly, the cancer cells treated with compounds were harvested for centrifugation at 1000 rpm for 5 min at room temperature, washed twice with PBS and resuspended with binding buffer, and then PI (Sigma, St. Louis, MO, USA) was added to a final concentration of 20 mg/mL. The cell lines were analyzed by flow cytometry (Becton Dickinson, Franklin Lakes, NJ, USA).

### 3.5. Statistical Analysis

The IC_50_ values were calculated from the semilogarithmic dose-response curves. The data were analyzed using SPSS 18.0 and reported as mean ± SD of the number of experiments indicated. For all measurements, one-way ANOVA followed by Student’s t-test was used to assess the statistical significance of the difference between each group. The LSD method was used to assess the statistical significance of the difference between the two groups. A statistically significant difference was considered at the level of *P* < 0.05. The data are presented as the mean ± SEM of three assays.

## 4. Conclusions

In this study, 27 novel L-shaped ortho-quinone analogs were synthesized and evaluated for their anti-cancer activities. Compounds **TB1**, **TB3**, **TB4**, **TB6**, **TC1**, **TC3**, **TC5**, **TC9**, **TC11**, **TC12**, **TC14**, **TC15**, **TC16**, and **TC17** possessed broad-spectrum potent cytotoxicity against PC3, K562, and WM9 cells. With more than a 70% inhibitory rate, **TB7** showed better inhibitory activity of K562 cells as compared with other cells. Moreover, we observed that **TC7** inhibited the growth of PC3 cells more efficiently than other cells. Some of the active compounds such as **TB3**, **TC1**, **TC3**, and **TC7** inhibited cell proliferation mainly through inducing apoptosis. The structure-activity relationships evaluation showed that removing methyl at C-3 of the furan ring and introducing diverse side chains at C-2 of the furan ring is an effective strategy for improving the anticancer activity of L-shaped ortho-quinone analogs.

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
