# Peer review of "Potent Cytotoxicity of Novel L-Shaped Ortho-Quinone Analogs through Inducing Apoptosis"

_molecules, 2019, doi:10.3390/molecules24224138_

Round 1

Reviewer 1 Report

Comments:

Page 1, line 6, “Chao Chen *, b, c,” should be amended “Chao Chen #, b, c,”? Page 10, line 374, “with a more than 70% inhibition rate; 5 μmol/L of TB7 and TC7 showed better inhibitory activity on K562 and PC3, respectively (Figure 2A).”, please authors confirm the inhibitory rate of the compound TB7 and TC7 in the Figure 2A. Page 14, line 410, “(Tables 1-2)” should be amended “(Tables 1 and 2)”. Page 16, line 462, “Scale bar = 100 μM in all images.” should be amended “Scale bar = 100 μm in all images.”? Page 19, line 480, “Compounds TB7 and TC7 displayed potent inhibitory activity against K562 and PC3 cells, respectively.”, please the authors to confirm the results in the conclusion. Page 19, line 494, “Int J Cancer 2019, 144(8), 1941-1953.” should be amended “Int J Cancer 2019, 144(8), 1941–”. Page 19, line 506, “Chem Res Chin Univ 2017, 33(1), 80―86.” should be amended “Chem Res Chin Univ 2017, 33(1), 80–”. Page 20, line 550, “Monatshefte für Chemie-Chemical Monthly 2015, 146(12): 2117-2126.” should be amended “Monatshefte für Chemie-Chemical Monthly 2015, 146(12): 2117–”. Page 20, line 553, “Molecules 2013, 18(8), 9818-9832.” should be amended “Molecules 2013, 18(8), 9818–”. Page 20, line 556, “Bioorg Med Chem 2016, 24(5), 1006-1013.” should be amended “Bioorg Med Chem 2016, 24(5), 1006–”. Page 20, line 564, “Bioorg Med Chem 2003, 11(14), 3179-3191.” should be amended “Bioorg Med Chem 2003, 11(14), 3179–3191”. Page 20, line 573, ” Chem Res. 2014, 23(11), 4631-4641. should be amended “Med Chem Res 2014, 23(11), 4631–4641.”.

Author Response

Dear Ms. reviewer,
     Thank you very much for your comments for my paper, the responses to reviewer's comments are as below:

Responses to reviewers’ comments

Comments1:

Page 1, line 6, “Chao Chen *, b, c," should be amended "Chao Chen #, b, c,"?

Answer:"Chao Chen *, b, c," was amended with "Chao Chen #, b, c,"(Page 1, line 6)

Page 10, line 374, "with a more than 70% inhibition rate; 5 μmol/L of TB7 and TC7 showed better inhibitory activity on K562 and PC3, respectively (Figure 2A).", please authors confirm the inhibitory rate of the compound TB7 and TC7 in the Figure 2A.(Page 10, line 374)

Answer: "with a more than 70% inhibition rate; 5 μmol/L of TB7 and TC7 showed better inhibitory activity on K562 and PC3, respectively (Figure 2A)." was amended with “with a more than 70% inhibition rate, TB7 showed better inhibitory activity on K562 cells compared with other cells. Moreover, we observed that TC7 inhibited the growth of PC3 cells more efficiently than other cells.” (Page 10, line 382)

3.Page 14, line 410, "(Tables 1-2)" should be amended "(Tables 1 and 2)".

Answer: "(Tables 1-2)" was amended with "(Tables 1 and 2)".(Page 14, line 414)

Page 16, line 462, "Scale bar = 100 μM in all images." should be amended "Scale bar = 100 µm in all images."?

Answer: "Scale bar = 100 μM in all images." was amended with "Scale bar = 100 µm in all images.".(Page 16, line 468)

Page 19, line 480, "Compounds TB7 and TC7 displayed potent inhibitory activity against K562 and PC3 cells, respectively.", please the authors to confirm the results in the conclusion.

Answer: "Compounds TB7 and TC7 displayed potent inhibitory activity against K562 and PC3 cells, respectively." was amended with “with a more than 70% inhibition rate, TB7 showed better inhibitory activity of K562 cells compared with other cells. Moreover, we observed that TC7 inhibited the growth of PC3 cells more efficiently than other cells”.(Page 19, line 484)

Page 19, line 494, "Int J Cancer 2019, 144(8), 1941-1953." should be amended "Int J Cancer 2019, 144, 1941–1953.".

Answer: "Int J Cancer 2019, 144(8), 1941-1953." was amended with "Int J Cancer 2019, 144, 1941–1953." (Page 19, line 499)

Page 19, line 506, "Chem Res Chin Univ 2017, 33(1), 80―86." should be amended "Chem Res Chin Univ 2017, 33, 80–86.".

Answer: "Chem Res Chin Univ 2017, 33(1), 80―86." was amended with "Chem Res Chin Univ 2017, 33, 80–86." (Page 19, line 512)

Page 20, line 550, "Monatshefte für Chemie-Chemical Monthly 2015, 146(12): 2117-2126." should be amended "Monatshefte für Chemie-Chemical Monthly 2015, 146, 2117–2126.".

Answer: "Monatshefte für Chemie-Chemical Monthly 2015, 146(12): 2117-2126." was amended with "Monatshefte für Chemie-Chemical Monthly 2015, 146, 2117–2126." (Page 20, line 557)

Page 20, line 553, "Molecules 2013, 18(8), 9818-9832." should be amended "Molecules 2013, 18, 9818–".

Answer: "Molecules 2013, 18(8), 9818-9832." was amended with "Molecules 2013, 18, 9818–9832." (Page 20, line 559)

Page 20, line 556, "Bioorg Med Chem 2016, 24(5), 1006-1013." should be amended "Bioorg Med Chem 2016, 24, 1006–1013.".

Answer: "Bioorg Med Chem 2016, 24(5), 1006-1013." was amended with "Bioorg Med Chem 2016, 24, 1006–1013." (Page 20, line 563)

Page 20, line 564, "Bioorg Med Chem 2003, 11(14), 3179-3191." should be amended "Bioorg Med Chem 2003, 11, 3179–3191.".

Answer: "Bioorg Med Chem 2003, 11(14), 3179-3191." was amended with "Bioorg Med Chem 2003, 11, 3179–3191." (Page 20, line 571)

Page 20, line 573, "Med. Chem Res. 2014, 23(11), 4631-4641." should be amended "Med. Chem. Res. 2014, 23, 4631–4641.".

Answer: "Med. Chem Res. 2014, 23(11), 4631-4641." was amended with "Med. Chem. Res. 2014, 23, 4631–4641.".(Page 20, line 580)

Reviewer 2 Report

I would recommend that the manuscript "Potent cytotoxicity of novel L-shaped ortho-quinone analogs through inducing apoptosis by damaging DNA integrity" be published in Molecules as long as the following points are addressed and the English grammatical mistakes are corrected:

1) Explain what "General cancer types" means, this is not the typical verbiage one would use to describe cancer

2) provide a reference after the fourth sentence of the introduction (line 38)

3) Don't abbreviate Beta-lapachone before introducing the term

4)Line 55, what is NQO1 and what L-shaped pocket are the authors talking about? What enzyme? What biological target?

5) Line 57 please explain what "the steric hinderance of the C-2 side chain is . . . more suitable for rotation" What rotation? this is a very rigid structure and I don't understand what the authors are talking about.

6) Figure 1 - please label the position numbers on the ring in question. This will help the readers orient to where the 2 position is. Also, compounds 4 and 5 are not even mentioned in the text. Please mention them or remove them from the figure.

7)There is no spectral data for compounds 7 and 8, please rectify this.

8) Assigned protons are missing from the following compounds: TC3, TC4, TC5, TC7, TC14, TC16, TC18. And the following compounds have an extra proton assigned to them: TC15

9) The following compounds are missing an assigned Carbon in their C13 spectra: TB2, TC2, TC13, TC17 and TC18. And the following compounds have an extra carbon assigned to them: TC6, TC12. Please rectify these mistakes

10) Line372, TC14 does not have broad spectrum inhibitory activity against all three cell lines

11) The inhibition data for paclitaxel was not listed for all three cell lines, so the authors can not make comparisons (line 381 and line 385)

12) Doesn't TC1 have a much better inhibitory activity against K562 than paclitaxel? I don't understand what the authors mean by "closed activity"

13) I have no idea what the authors mean by ""introduction of alkane groups to C2 of furan ring of compound 9" (lines 413 and 418). What is compound 9?

Please have all of these items addressed before publication.

Author Response

Dear Ms. reviewer,
     Thank you very much for your comments for my paper, the responses to reviewer's comments are in word.

Reviewer 3 Report

The manuscript reports the chemical synthesis and cytotoxicity of several novel L-shaped ortho-quinone analogs. While the information provided on the chemical synthesis and structural characterization of the analogs are sufficiently described, biological data on cytotoxic effects need a revision.

Cytoxicity at a fixed concentration was initially tested by MTT assay. Figure 2A and table 1 report the same data and the first one can be removed. In this case, table 1 can be modified highlighting those compounds which activity is a above a threshold value (i.e. 70% toxicity).

Again figure 2B and table 2 reports the same data. In addition, letters and symbols in figure 2 are too small to be easily read and there are some discrepancies between data reported. In particular, TC9 compound seem to be the less effective on PC3 cell line (figure 2B) while is reported as one with the strongest activity in table 2. Which is correct?

Data on cell apoptosis and DNA damage need to be revised. Microscopic observation reported in figure 3 cannot be use to asses apoptosis. Reduction in cell number can be monitored with more appropriate techniques (i.e. cell counting by cytometry using tripan blue as dye) while the evaluation of changes in cell morphology need a better analysis.

Cell apoptosis evaluated by annexin V staining lacks a positive control (i.e. camptotecin) and should be associated with an additional test (i.e. caspase activation, PARP cleavage). Moreover, DNA damaging is not supported by the reported data. It is impossible to see a clear ladder profile in the gel images, but just a smear profile. No positive control is provided. DNA damage can be monitored using different assays (H2AX phosphorylation, TUNNEL, comet assay, oxoG and so on). In my opinion the conclusion that tested compounds can induce apoptosis by damaging DNA integrity opinion are not supported by provided data, then more information and better evidences are required.

Author Response

Dear Ms. reviewer,
     Thank you very much for your comments for my paper, the responses to reviewer's comments as below:

Responses to reviewers’ comments

Comments 3:

Cytoxicity at a fixed concentration was initially tested by MTT assay. Figure 2A and table 1 report the same data and the first one can be removed. In this case, table 1 can be modified highlighting those compounds which activity is a above a threshold value (i.e. 70% toxicity).

Answer: according to your suggestion, we have deleted the Figure 2A. (page 11, line 395)

Again figure 2B and table 2 reports the same data. In addition, letters and symbols in figure 2 are too small to be easily read and there are some discrepancies between data reported. In particular, TC9 compound seem to be the less effective on PC3 cell line (figure 2B) while is reported as one with the strongest activity in table 2. Which is correct?

Answer: we have checked the data and corrected the data. (page 11, line 398)

Data on cell apoptosis and DNA damage need to be revised. Microscopic observation reported in figure 3 cannot be use to asses apoptosis. Reduction in cell number can be monitored with more appropriate techniques (i.e. cell counting by cytometry using tripan blue as dye) while the evaluation of changes in cell morphology need a better analysis.

Answer: in the study of cell apoptosis, we first found that some cells had apoptotic bodies according to the microscopic observation, but the apoptosis was not confirmed according to the microscopic observation. Then flow cytometer detection was used to verify that a certain degree of apoptosis did occur. DNA damage is just a targeted test in this paper, and we found that through three tests, DNA damage occurred in cells that did have partial compound treatment, and a lot of DNA fragments of different sizes appeared, the specific molecular mechanism of which we are now studying. (page 14, line 427)

Cell apoptosis evaluated by annexin V staining lacks a positive control (i.e. camptotecin) and should be associated with an additional test (i.e. caspase activation, PARP cleavage). Moreover, DNA damaging is not supported by the reported data. It is impossible to see a clear ladder profile in the gel images, but just a smear profile. No positive control is provided. DNA damage can be monitored using different assays (H2AX phosphorylation, TUNNEL, comet assay, oxoG and so on). In my opinion the conclusion that tested compounds can induce apoptosis by damaging DNA integrity opinion are not supported by provided data, then more information and better evidences are required.

Answer: thanks for your advice. We are now further in-depth studying DNA damage and repair gene regulation and its molecular mechanism of the active compounds, independent research papers will be presented further added to explain the experiment results, but we did detect many different sizes of DNA fragments in the cancer cells by treating with some active compound using DNA ladder, therefore, we concluded that the active compounds can cause DNA damage. (page 14, line 460)

Responses to reviewers’ comments

Round 2

Reviewer 1 Report

Comments:

Page 10, line 377, and Page 18, line 478, “TB7” should be amended “TB7”. Page 18, line 479, “TC7” should be amended “TC7”? Page 20, line 571, ” Med. Chem. Res.” should be amended “Med Chem Res”.

Author Response

Cover letter

Dear Editor:

On behalf of my co-authors, I am submitting the revised manuscript “Potent cytotoxicity of novel L-shaped ortho-quinone analogs through inducing apoptosis by damaging DNA integrity” for possible publication in Molecules.

We certify that we have participated sufficiently in the work to take public responsibility for the appropriateness of the experimental design and method, and the collection, analysis, and interpretation of the data.

We have reviewed the present version of the manuscript and approve it for publication.

Yours sincerely,

Prof./Dr. Weidong PAN

11/11/2019

3491 Baijin Road, Guiyang 550014, PR China

Corresponding author: Chao Chen, [email protected],

Heng Luo, [email protected],

                  Wei-Dong Pan, [email protected].

Responses to reviewers’ comments

Question: “TB7”should be amended “TB7”. Page 18, line 479, “TC7” should be amended “TC7” (Page 11, line 382 and 383. Page 18, line 483 and 484.)  

Answer: “TB7”, “TC7” was amended with “TB7”, “TC7” respectively.

Question: “ Chem. Res.”should be amended “Med Chem Res”. (Page 20, line 580), ”

Answer: The reference was deleted

Page 18, line 491, We added a project number “QKHJC [2019]2757”.

Question: “Authors modified in some way the manuscript but still remain my concern about the conclusions drawn. DNA damaging is based only on the presence of some lower molecular weight fragments (in absence of markers is impossible to get an idea of their size). Their presence can be attributed to several causes including the extraction procedure and/or the activation of nucleases and not directly to a chemical DNA damage incuded by the  treatment with the analyzed compounds. In addition, no information are provided about the amount of DNA loaded (is it the same for each line?, how it has been measured?, samples were not treated with RNAse). So differences in the presence/absence of lower size fragments can be the result of an uneven amount of loaded material as well as it is possible a RNA contamination which will result in a smear of fragments at lower sizes. Authors state that “the molecular mechanism needs further study” (sentence in line 454 need to be rephrased since is incomplete) but they claim that a DNA damage is induced by some compounds. Still in my opinion DNA damage should be assessed in a more precise manner.

Anyway sentences in lines 453-454 “The results showed that DNA integrity may be damaged by treating with TB1, TC1, TC3, TC7, TC17, TC11 and TC14, while TB6 and TC9 showed no significant damage effects” and lines 408-481 “TB3, TC1, TC3, TC7 inhibited cell proliferation may be mainly through inducing apoptosis by damaging DNA integrity.” do not fit with the reported data on “DNA fragmentation”. While it is possible to observe an higher quantity of low size fragments in the case of treatment with TB1 and TC1, a less evident profile for TC3 and TC7, no data is reported for TC14 (claimed as DNA damage inducing agents), poor or no evidence are for TC11 and TC17.

Conclusions drawn and part of the manuscript title “….inducing apoptosis by damaging DNA integrity” are still not supported by clear evidences.”(Page 1, line 24, Page 9, line338, Page14, line443, Page 17, line 459, 466)

  Answer: In order to respect the opinions of the reviewers, we decided to delete the results of DNA ladder research and also revised the corresponding research conclusions of paper. At the same time, I am very grateful that I received the valuable comments of reviewers. Now our research group is conducting in-depth systematic research on the DNA damage of active compounds and its mechanism. The corresponding research results will be reported in another article. And we added a sentence “Some of the active compounds such as TB3, TC1, TC3, TC7 inhibited cell proliferation mainly through inducing apoptosis”(Page 17, line 466).

Page 1, line 27, “β-lapachone” was amended with “Beta-lapachone”.

Page 1, line 32, “The general cancer types in Chinese male, in 2018, were lung, stomach, colorectum, liver and esophageal cancer.” was amended with “The common cancer types in Chinese male, in 2018, were lung, stomach, colorectum, liver and esophageal cancer.”.

Page 1, line 48, “Furthermore, β-lap is another natural ortho-quinone, which has been reported to kill many human cancer cells selectively[14]. However, It has been found that the pyran ring of β-lap was unstable during the metabolic process of human body, and may led to side effects to normal tissues[15,16]. Recently, studies revealed that some tanshinone analogs showed similar or stronger antitumor activity when removing the ring-A but retaining the furan ring[17,18]. In addition, You et al.[19,20] found that the binding site for NQO1 substrates was an L-shaped pocket, which binds well with L-shaped molecular and showed higher antitumor activities than the planar compound 1 and β-lapachone. However, the steric hindrance of the C-2 side chain is small without methyl at C-3 of furan ring and more suitable for rotation. So we anticipate that a novel L-shaped molecule without methyl at C-3 of furan ring could provide better antitumor activities. ” was amended with “Furthermore, Beta-lapachone is another natural ortho-quinone, which has been reported to kill many human cancer cells selectively[14]. However, It has been found that the pyran ring of beta-lapachone was unstable during the metabolic process of human body, and may led to side effects to normal tissues[15,16]. Recently, studies revealed that some tanshinone analogs showed similar or stronger antitumor activity when removing the ring-A but retaining the furan ring[17,18]. In addition, You et al.[19,20] found that the binding site for quinone oxidoreductase-1 (NQO1) substrates was an L-shaped pocket(Figure 1 B), which binds well with tanshinone analogs and showed higher antitumor activities than the planar compound 1 and beta-lapachone. Therefore, we surmise that removing methyl at C-3 of furan ring is more suitable for the binding site. And we anticipate that a novel L-shaped molecule without methyl at C-3 of furan ring could provide better antitumor activities. Considering that some nitrogen, oxygen-substituted and amino acid substrates may improve aqueous solubility and antitumor activities”.

Page 3, line 69, We added a sentence “Figure 1. (A) Structural design strategy. (B) L-shaped pocket: The description could get from reference[19]”. Page 3, line 89 and line 96, We added a sentence “Other data could be found in reference [26].” Page 7, line 268, “13C NMR (150 MHz, DMSO-d6) δ86, 174.62, 159.03, 157.32, 155.16 (d, J= 231.0 Hz), 145.06, 135.47, 130.37, 129.85, 129.69, 128.27, 122.42, 122.06, 115.84, 115.70, 113.90, 113.86, 105.93.” was amended with “13C NMR (150 MHz, DMSO-d6) δ 179.86, 174.62, 159.03, 157.32, 155.16 (d, J = 231.0 Hz), 145.06, 135.47, 130.37, 129.85, 129.69, 128.27, 122.42, 122.06, 115.77(d, J = 21.0 Hz), 113.88 (d, J = 6.0 Hz), 105.93.”. Page 10, line 369, “with a more than 70% inhibition rate;5 μmol/L of TB7 and TC7 showed better inhibitory activity on K562 and PC3, respectively (Figure 2A).” was amended with “with a more than 70% inhibition rate, TB7 showed better inhibitory activity on K562 cells compared with other cells. Moreover, we observed that TC7 inhibited the growth of PC3 cells more efficiently than other cells.” Page 10, line 376, “and another active compounds exhibited closedactivity to that of the positive control. The inhibition activity of TC1 against the growth of K562 was closed to that of the positive control, Paclitaxel, and better than that of another positive control, tanshinone ⅡA” was amended with “and another active compounds exhibited similar activity to that of the positive control. The inhibition activity of TC1 against the growth of K562 was better than that of the positive control, Paclitaxel and tanshinone ⅡA” Page 11, line 385, “Inhibition of L-shaped ortho-quinone analogs on the proliferation of cancer cell lines in vitro.(A) Cancer cell lines PC3, K562 and WM9 were incubated with quinone analogs at a final concentration of 5 μmol/L and the effect on cell proliferation was investigated. (B) Growth inhibition induced by the active compounds on PC3, K562 and WM9 cells by MTT assay.” was amended with “Growth inhibition induced by the active L-shaped ortho-quinone analogs on PC3, K562 and WM9 cells by MTT assay.” Page 12, line 392, “4528.039±” was amended with “72.452±8.039” Page 13, line 392and line 394, Due to our negligence We forgot to add some date in Table 1, Now It was added in Table 1. Page 13, line 403, “Whereas the introduction ofalkane groups to C-2 of furan ring of compound 9 resulted in decreased cytotoxicity in PC3 cell lines, K562 cell lines and WM9 cell lines. such as compounds  The TC series molecules with electron-withdrawing groups, saturate six-membered rings or multi-substituted groups emerged greater inhibitory effects on three cancer cell lines, such as TC11, TC12, TC15 and TC16. Whereas the introduction of donating groups or alkane groups to C-2 of furan ring of compound 9 resulted in reduced cytotoxicity in three cancer cell lines. The structure-activity relationships evaluation also showed that removing methyl at C-3 of furan ring and introducing diverse side chain at C-2 of furan ring is an good strategy for improving the anticancer activity of L-shaped ortho-quinone analogs.” was amended with “Whereas the TB series molecules bearing alkane groups at 2-position showed decreased cytotoxicity in PC3 cell lines, K562 cell lines and WM9 cell lines. such as compounds TB9. The TC series molecules with electron-withdrawing groups, saturate six-membered rings or multi-substituted groups emerged greater inhibitory effects on three cancer cell lines, such as TC11, TC12, TC15 and TC16. Whereas the TC series molecules bearing donating groups or alkane groups at 2-position showed reduced cytotoxicity in three cancer cell lines. The structure-activity relationships evaluation also showed that removing methyl at C-3 of furan ring and introducing diverse side chain at C-2 of furan ring are a good strategy for improving the anticancer activity of L-shaped ortho-quinone analogs.” Page 17, line 479, “Compounds TB7and TC7 displayed potent inhibitory activity against K562 and PC3 cells, respectively. TB3, TC1, TC3, TC7 inhibited cell proliferation mainly through inducing apoptosis by damaging DNA integrity.” was amended with “with a more than 70% inhibition rate, TB7 showed better inhibitory activity of K562 cells compared with other cells. Moreover, we observed that TC7 inhibited the growth of PC3 cells more efficiently than other cells.

Reviewer 3 Report

Authors modified in some way the manuscript but still remain my concern about the conclusions drawn. DNA damaging is based only on the presence of some lower molecular weight fragments (in absence of markers is impossible to get an idea of their size). Their presence can be attributed to several causes including the extraction procedure and/or the activation of nucleases and not directly to a chemical DNA damage incuded by the  treatment with the analyzed compounds. In addition, no information are provided about the amount of DNA loaded (is it the same for each line?, how it has been measured?, samples were not treated with RNAse). So differences in the presence/absence of lower size fragments can be the result of an uneven amount of loaded material as well as it is possible a RNA contamination which will result in a smear of fragments at lower sizes. Authors state that “the molecular mechanism needs further study” (sentence in line 454 need to be rephrased since is incomplete) but they claim that a DNA damage is induced by some compounds. Still in my opinion DNA damage should be assessed in a more precise manner.

Anyway sentences in lines 453-454 “The results showed that DNA integrity may be damaged by treating with TB1, TC1, TC3, TC7, TC17, TC11 and TC14, while TB6 and TC9 showed no significant damage effects” and lines 408-481 “TB3, TC1, TC3, TC7 inhibited cell proliferation may be mainly through inducing apoptosis by damaging DNA integrity.” do not fit with the reported data on “DNA fragmentation”. While it is possible to observe an higher quantity of low size fragments in the case of treatment with TB1 and TC1, a less evident profile for TC3 and TC7, no data is reported for TC14 (claimed as DNA damage inducing agents), poor or no evidence are for TC11 and TC17.

Conclusions drawn and part of the manuscript title “….inducing apoptosis by damaging DNA integrity” are still not supported by clear evidences.

Author Response

(The authors gave the same response as above.)
